# Intra-Patient Lung CT Registration through Large Deformation Decomposition and Attention-Guided Refinement

**DOI:** 10.3390/bioengineering10050562

**Published:** 2023-05-08

**Authors:** Jing Zou, Jia Liu, Kup-Sze Choi, Jing Qin

**Affiliations:** 1Center for Smart Health, School of Nursing, the Hong Kong Polytechnic University, Hong Kong, China; jing9.zou@connect.polyu.hk (J.Z.); jia.liu@siat.ac.cn (J.L.);; 2Shenzhen Institutes of Advanced Technology, Chinese Academy of Sciences, Shenzhen 518055, China

**Keywords:** deformable medical image registration, intra-patient, lung CT, deformation field decomposition, attention layer refinement

## Abstract

Deformable lung CT image registration is an essential task for computer-assisted interventions and other clinical applications, especially when organ motion is involved. While deep-learning-based image registration methods have recently achieved promising results by inferring deformation fields in an end-to-end manner, large and irregular deformations caused by organ motion still pose a significant challenge. In this paper, we present a method for registering lung CT images that is tailored to the specific patient being imaged. To address the challenge of large deformations between the source and target images, we break the deformation down into multiple continuous intermediate fields. These fields are then combined to create a spatio-temporal motion field. We further refine this field using a self-attention layer that aggregates information along motion trajectories. By leveraging temporal information from a respiratory cycle, our proposed methods can generate intermediate images that facilitate image-guided tumor tracking. We evaluated our approach extensively on a public dataset, and our numerical and visual results demonstrate the effectiveness of the proposed method.

## 1. Introduction

Deformable medical image registration (DMIR) is a critical task in various clinical applications, including surgery planning [1], diagnosis [2], radiation therapy [3,4], and longitudinal studies [5]. It involves aligning multiple images acquired at different times, from different patients, or from different positions. Among various medical imaging registration challenges, lung CT registration is particularly difficult due to the large and non-rigid deformations caused by organ motion, respiratory cycles, and pathological changes [6]. The accurate and efficient registration of lung CT images is crucial for disease diagnosis, monitoring, and radiation therapy planning. Therefore, developing a patient-specific lung CT image registration method that addresses these challenges is of paramount importance.

The registration process involves finding a transformation that aligns the anatomical structures in a source image with those in a target image. In early studies, traditional registration methods have been proposed, such as Large Diffeomorphic Distance Metric Mapping (LDDMM) [7,8,9] and symmetric methods [10,11,12]. However, these traditional methods are often computationally expensive and time-consuming as they formulate the registration problem as an independent pair-wise iterative optimization problem. The registration process becomes significantly time-consuming when dealing with image pairs containing significant deformations in anatomical structures. This limitation prevents such techniques from being used in many clinical scenarios where real-time registration is required. Moreover, these methods often struggle to handle large and complex deformations between images, such as those caused by organ motion or respiration.

Deep learning techniques have been utilized to solve deformable registration problems in recent years. Supervised methods were the first to be introduced for performing image registration, where a ground truth deformation field is necessary for training. However, these ground truth deformation fields are difficult to obtain, especially for medical image registration tasks. Then, researchers turned to unsupervised methods and semi-supervised methods, which eliminated the requirement of ground truth. Many unsupervised methods and semi-supervised methods have achieved remarkable success [13,14,15,16,17,18,19,20].

Despite the progress made in the use of deep learning techniques for deformable image registration, accurately estimating large and irregular deformations caused by respiration remains a challenging task. This is particularly important in the context of medical image registration, as tumors and sensitive structures in the thorax can move more than 20 mm [21] during respiration, making it difficult to achieve accurate registration. Figure 1 shows a large deformation in the diaphragm and near the heart area.

We propose a patient-specific method for deformable lung CT image registration by decomposing the large deformation fields into several continuous intermediate fields. This approach allows us to better capture the complex and large deformations that occur in the lung and improves the accuracy of the registration results. Our proposed approach takes advantage of the temporal characteristics within a respiratory cycle to create intermediate images, which can be valuable in applications such as tumor tracking within image-guided systems. During the training process, both the extreme phase images and the intermediate images are utilized as training data. After the network has completed the training process, it is capable of estimating a deformation field directly without relying on any intermediary images.

We conduct an evaluation of our proposed method on a publicly available dataset, and the results of our experiments show that our approach is effective in enhancing the accuracy and efficiency of lung CT registration. The proposed method has potential applications in radiation therapy planning, disease monitoring, and other clinical scenarios that require accurate and efficient lung CT registration.

## 2. Related Works

### 2.1. Pair-Wise Optmization

Traditional image registration methods have been widely studied in medical image registration. These methods usually treat registration as an independent pair-wise iterative optimization problem, where the objective is to find the optimal transformation parameters that map one image to another. Many toolkits are open to the public for medical image analysis, such as Advanced Normalization Tools (ANTs) [22], Simple Elastix [23], and NeftyReg [24]. One of the most well-known traditional methods is Large Diffeomorphic Distance Metric Mapping (LDDMM) [7,8,9]. The registration process is represented as a partial differential equation that governs the deformation field between two images and describes the transformation that aligns one image with another. LDDMM is able to handle large deformations and topological changes in the image, making it suitable for registration tasks in medical imaging. Another example is Symmetric Normalization (SyN) [10,11,12], which uses a symmetric diffeomorphic transformation to optimize the similarity between two images.

However, these traditional methods have several limitations. One of the major drawbacks is that they are usually computationally expensive and time-consuming, and they are particularly problematic when dealing with image pairs that exhibit significant deformations in anatomical structures. This is because they need to solve a complex optimization problem for each pair of images independently. Moreover, traditional methods often require manual initialization, which is labor-intensive and time-consuming. Another issue is that traditional methods are sensitive to the choice of similarity measures and regularization terms, which require expert knowledge to tune properly. Recent studies have addressed the aforementioned limitations by focusing on the development of deep-learning-based methods for medical image registration. These methods aim to directly learn the deformation field from the input images, eliminating the need for explicit optimization.

### 2.2. Learing-Based Registration

In recent years, image registration methods based on machine learning have gained popularity. These approaches can be broadly classified into three categories based on the degree of supervision required: supervised, unsupervised, and weakly supervised methods.

Supervised methods for image registration [25,26,27] rely on ground truth deformation fields, which can be either real or synthetic. The real deformation fields are obtained through traditional registration techniques, while synthetic deformation fields are generated using statistical models or random transformations. These fields are used as a reference to train the registration algorithm. The quality and quantity of the ground truth fields significantly affect the performance of the model.

Unsupervised methods for image registration do not require annotated training data and instead learn the deformation fields by minimizing the dissimilarity between the target and transformed source images. Balakrishnan et al. [20] proposed an unsupervised method and adopted a Spatial Transformer Network (STN) [28] for image transformation. De Vos et al. [14] presented an unsupervised multi-stage network for affine and deformable registration adopting a coarse-to-fine strategy, while CycleMorph with cycle-consistency [17], inverse-consistency [29,30], and symmetric pyramid network [31] are proposed to improve the registration performance. However, these methods are trained in a bi-directional manner, which highly increases the computational complexity and training time.

The semi-supervised networks [32,33] leverage the mask or landmark information as supervisions during the training. In a popular network, VoxelMorph [13], a semi-supervised strategy, was utilized by supervising the learning of the deformation field with brain masks. For lung CT image registration, the network in [16] leverages lung masks and landmarks for semi-supervised learning. However, in the medical image domain, acquiring masks and labels is challenging and costly.

### 2.3. Self-Attention

Self-attention [34] is a mechanism in deep learning that enables a network to selectively focus on different parts of an input sequence or image, allowing it to encode relevant information and discard irrelevant information. Several studies have explored the use of self-attention in medical image registration, which has shown promising results.

Chen et al. [35] introduced a novel architecture for medical image registration, named ViT-V-Net, which combines ConvNet and Transformer models for brain MRI registration. They incorporated the Vision Transformer (ViT) [36] to learn high-level features, and found that replacing the VoxelMorph backbone with ViT-V-Net improved the registration accuracy. They further extended ViT-V-Net to TransMorph [37], a hybrid Transformer-ConvNet framework that uses the Swin Transformer [38] as the encoder to learn spatial transformations between input images. After processing the data from the Transformer encoder, a decoder built with ConvNet generated the dense deformation field. In another work, Liu et al. [39] proposed multi-organ PET and CT multi-modality registration based on a pre-trained 2D Transformer model.

One advantage of using self-attention in medical image registration is its ability to capture global dependencies and long-range interactions, which are particularly relevant in medical imaging where structures can undergo large deformations. Additionally, self-attention can capture complex spatial relationships between different regions of the image, improving the accuracy of the registration. However, one disadvantage of self-attention-based registration methods is their relatively high computational cost. Self-attention requires the computation of a pairwise similarity matrix between all feature vectors in the input sequence, which can be computationally expensive for large input sizes. In this work, we alleviate this disadvantage by utilizing only one self-attention layer for spatial-temporal field refinement, rather than using the whole Transformer architecture [34].

## 3. Methodology

### 3.1. Problem Statement

Our objective is to obtain the deformation field $\Phi \in R^{H\times W\times C\times 3}$ that represents the spatial transformation between the source lung CT image $I_{S}$ and the target lung CT image $I_{T}$, where $I_{S}$ and $I_{T}$ are H×W×C tensors. To achieve this, we utilize a deep neural network *f*, which takes $I_{S}$ and $I_{T}$ as inputs and generates Φ. The goal is to solve the following problem:(1)$$\begin{array}{c} \underset{f\in F}{\mathrm{argmin}}\ell \left(I_{T},I_{S}\circ \Phi_{f}\right)+\lambda R\left(\Phi_{f}\right), \end{array}$$
where F represents the function space of *f* and $\Phi_{f}$ represents the deformation field obtained using *f* for the given inputs. The term $I_{S}\circ \Phi_{f}$ represents the warped image obtained by applying $\Phi_{f}$ on $I_{S}$. The loss function *ℓ* is used to measure the difference between $I_{T}$ and $I_{S}\circ \Phi_{f}$. Additionally, we introduce a regularization term $R(\Phi_{f})$ with a hyperparameter λ to control its influence on the optimization.

While the current training approach is effective for small deformations in lung CT images [7], it fails to produce accurate results when dealing with large and irregular deformations where pixels undergo significant transformations. To address this limitation, our method decomposes the deformation field into several small ones and gradually improves them using an attention layer. Figure 2 provides an overview of our proposed approach.

### 3.2. Decomposition

To address the issue of large and irregular deformations in lung CT image registration, we propose a method that decomposes the deformation field Φ into multiple intermediate fields $u_{i}$, which describe the direction and distance of voxel movement from $I_{S}$ to $I_{T}$. We assume that the movement of each voxel is along a smooth line, and therefore, we use linear interpolation to obtain the intermediate fields $u_{i}$ by incrementally moving the voxels along the deformation path.
(2)$$\begin{array}{c} u_{i}=\Phi /n, \end{array}$$*n* represents the number of phases in a respiratory cycle.

### 3.3. Refinement

The linear interpolation used for decomposing the deformation field assumes a homogeneous displacement of each voxel, which may not always hold true in practical scenarios. To address this, we refine the small deformation fields by concatenating them to create a spatio-temporal motion field *U* that contains both spatial and temporal information of the respiratory motion. We then pass this motion field *U* through a self-attention layer to obtain a refined field *V*, which is decomposed again to generate refined intermediate fields $v_{i}$. Using these refined fields, we warp the source image $I_{S}$ to generate intermediate images $I_{n}$, which are compared with ground truth intermediate images $T_{n}$ to calculate the loss.

#### Attention Layer

Self-attention is a mechanism commonly used in deep learning models that enables the model to selectively weigh the importance of different parts of the input sequence in making predictions. It achieves this by calculating the dot product of a query matrix with a key matrix, applying a softmax activation function to the resulting scores, and then multiplying the output by a value matrix to obtain the final representation. This allows the model to capture global information from the input sequence in a more efficient and effective manner. The self-attention formula is given as follows:(3)$$\mathrm{Attention}\left(Q,K,V\right)=\mathrm{softmax}\left(\frac{QK^{T}}{\sqrt{d_{k}}}\right)V,$$
where *Q*, *K*, and *V* are matrices representing query, key, and value, respectively, with dimensions $n\times d_{k}$, where *n* is the sequence length and $d_{k}$ is the dimensionality of the key and value vectors. The dot product $QK^{T}$ is divided by $\sqrt{d_{k}}$ to mitigate the effects of vanishing gradients. Finally, the softmax function is applied to the scores to obtain attention weights, which are then used to weight the value vectors *V* to obtain the final output.

In our experiment, we construct an attention layer consist of Layer Norm (LN) operations [40] and residual connections [41]:(4)$$\begin{array}{c} y=y+Norm(Attention(x)), \end{array}$$

### 3.4. Network Update

In our experiments, we used Mean Square Error (MSE) as the similarity matrix. This metric calculates the average of the squared differences between the target image and the transformed source image. The MSE is computed by taking the sum of the squared residuals and dividing it by the number of voxels in the images. Mathematically, it can be expressed as follows:(5)$$\ell =\frac{1}{\left|\Omega \right|}\sum_{p\in \Omega }{\left(I_{T}\left(p\right)-I_{S}\circ \left(\Phi \right)\left(p\right)\right)}^{2}$$

Here, $I_{T}\left(p\right)$ and $I_{S}\circ \left(\Phi \right)\left(p\right)$ represent the intensity values of the target image and the deformed source image, respectively, at location *p*. |Ω| is the total number of voxels in the image domain Ω,

To ensure the smoothness and physical plausibility of the deformation field Φ, we apply regularization during training. A diffusion regularization term is imposed on the spatial gradients of the deformation field to penalize large displacements. The regularization term can be expressed as follows:(6)$$R\left(\Phi \right)=\sum_{p\in \Omega }{\left||\nabla (\Phi (p)|\right|}^{2},$$
where $\left|\right|\cdot {\left|\right|}^{2}$ denotes the L2 norm of the vector. We use finite differences to approximate spatial gradients, such that $\frac{\partial (\Phi )(p)}{\partial x}\approx \left(\Phi \right)\left(\left(p_{x}+1,p_{y},p_{z}\right)\right)-\left(\Phi \right)\left(\left(p_{x},p_{y},p_{z}\right)\right)$ and similarly for $\frac{\partial u(p)}{\partial y}$ and $\frac{\partial (\Phi )(p)}{\partial z}$. This regularization term encourages a smoother deformation field Φ that is more physically plausible.

The ultimate goal of our decomposition approach is to train the deep neural network *f* to perform deformable registration. To achieve this, we aim to solve the following optimization problem:(7)$$\begin{array}{c} \underset{f\in F}{\mathrm{argmin}}\sum_{t=1}^{n}\ell_{t}\left(T_{t},I_{S}\circ \left(t\Phi_{f}/n\right)\right)+\lambda R\left(\Phi_{f}\right). \end{array}$$
where $\ell_{t}$ represents the different loss function at the decomposed stage and $T_{t}$ is the target intermediate images.

## 4. Experiments

### 4.1. Dataset

DirLab 4DCT dataset [42] contains 100 lung CT volumes collected from 10 patients, each having 10 CT volumes, denoted by $T_{0}$, ⋯, $T_{9}$, respectively, sampled at 10 different respiratory phases during a whole respiration period. All images have an isotropic in-plane resolution of 256 × 256 from 4DCT1 to 4DCT5 and of 512 × 512 from 4DCT6 to 4DCT10. The axial slices were acquired with a voxel size ranging from 0.97 mm to 1.98 mm, and the slice thickness and increment were 2.5 mm. This dataset also provides 300 anatomical landmarks for the volumes of $T_{0}$ and $T_{5}$, representing the maximal inspiration and the maximal expiration, and 75 landmarks for the remaining volumes; these landmarks are useful for evaluating the registration performance. For additional information regarding the dataset used in our study, please refer to the provided link for the dataset (the dataset is released at the website https://med.emory.edu/departments/radiation-oncology/research-laboratories/deformable-image-registration/downloads-and-reference-data/4dct.html, accessed on 8 May 2023).

### 4.2. Data Pre-Processing

The data pre-process includes resampling, affine registration, and cropping. Firstly, all images are re-sampled into a voxel spacing of 1 mm × 1 mm × 2.5 mm. Then we performed affine registration for all cases by affine registering $T_{5}$ to $T_{0}$, because in this study we focus on deformable registration. In the end, all the images are resized to have the same volume size of 240 × 160 × 96.

### 4.3. Performance Metrics

We utilized two metrics to evaluate our registration algorithm: (1) Target Registration Error (TRE) and (2) the percentage of negative Jacobian Determinant values.

#### 4.3.1. Target Registration Error (TRE)

The Target Registration Error (TRE) is used as the primary metric to assess the performance of our algorithm. TRE is a widely-used performance metric for landmark-based registration tasks, which measures the average Euclidean distance between landmarks in the target image and the corresponding landmarks in the registration output. A smaller TRE value indicates better registration performance.
(8)$$\begin{array}{c} TRE=\frac{1}{num}\sum_{i=1}^{num}\sqrt{{\left((p-q)\cdot s\right)}^{2}}, \end{array}$$
where num is the total number of the landmarks used to evaluate the registration algorithms. $p\in R^{3}$ is the landmark in the source image and $q\in R^{3}$ is the landmark in the target image after transformation. *s* is the spacing of the images; in our experiments, the value is [1.0, 1.0, 2.5].

#### 4.3.2. Jacobian Determinant

The Jacobian determinant is a measurement of the local spatial distortion of an image transformation. It quantifies how much the transformation expands or contracts the local volume of tissue. The calculation of the Jacobian determinant involves finding the partial derivatives of the transformation function with respect to each dimension of the image.
(9)$$\begin{array}{c} det\left(J\left(i,j,k\right)\right)=\left|\begin{array}{ccc} \frac{\partial i}{\partial x} & \frac{\partial j}{\partial x} & \frac{\partial k}{\partial x}\\ \frac{\partial i}{\partial y} & \frac{\partial j}{\partial y} & \frac{\partial k}{\partial y}\\ \frac{\partial i}{\partial z} & \frac{\partial j}{\partial z} & \frac{\partial k}{\partial z} \end{array}\right| \end{array}$$

The determinant can be negative, positive, or zero. If the determinant of the Jacobian matrix is equal to 1, then there is no volume change during the transformation. If it is greater than 1, it means the volume has expanded, and if it is between 0 and 1, it means the volume has shrunk. However, a determinant less than or equal to 0 suggests that there is a singularity in the image after registration, which is a point where the image has been folded or distorted. By calculating the percentage of negative determinant values, we can quantify the quality of the deformation field.

### 4.4. Implementation Details

The algorithm is implemented with pytorch lightning framework on an RTX 3090 GPU. We employed 3D UNet [43] as the backbone network due to its high performance. We ran 600 epochs with the batch size of 1 which is set according to the memory of our GPU card. The hyperparameter before the regularization term in the loss function is set as 0.01. The optimizer we used is Adam [44] with an initial learning rate of 0.0001.

### 4.5. Experimental Results

#### 4.5.1. Performance Improvement

We evaluate the performance of our proposed method by comparing it with five existing methods, namely BL [20], IL [14], VM [13], MAC [16], and CM [17], denoting the baseline and existing methods that use the iterative learning strategy, lung masks as the supervision, landmarks as the supervision, and the cycle consistency, respectively. To ensure a fair comparison, we adopt the same backbone network (3D UNet) and learning settings for all methods.

The results of our experiments are presented in Table 1 obtained via cross-validation. Our proposed method achieved the best performance in terms of average target registration error (*Avg.* in the table) compared to the other five competitive methods. Specifically, our approach outperformed the second-best method (VM) by 8.0%, indicating its superior effectiveness. Moreover, our method yielded the best results for five out of the ten patients, and it also exhibited the lowest standard deviation (*Std.* in the table), indicating its reliability. These results demonstrate the effectiveness and robustness of our method.

#### 4.5.2. Statistical Analysis

We conducted a statistical significance test to confirm the improvement in performance using a non-parametric test—the Friedman test for multiple comparisons. The statistical analysis for the 4DCT dataset is reported in Figure 3. We can conclude that there is a statistically significant difference in performance compared to other methods with $\chi_{Friedman}^{2}\left(5\right)=48.37$ and $p=2.2\times 10^{-7}$. In the figure, we also observed an outlier on the results using the MAC algorithm [16]. We do not find an outlier in our method, which demonstrates the effectiveness of our method again.

#### 4.5.3. Evaluation on Jacobian Determinant

The percent of negative values of the Jacobian determinant is shown in Table 2. We can see that our method achieved the lowest percent (0.016) compared to other methods. Negative determinants indicate there is distortion or folding in the deformation field. Thus, the registration results of our method show less abrupt changes and outperform the others in terms of keeping the smoothness of the deformation field during registration.

#### 4.5.4. Qualitative Evaluation

The visualization of results is shown in Figure 4. The first row displays the difference image before registration obtained by subtracting the target image from the source image. We can see that there are large differences in the diaphragm and near the heart area, which are illustrated as highlighted black areas. The second row shows the difference images after registration; we can observe that after registration, the large differences are all eliminated. The last row presents the deformation field from three different perspectives. In these images, the large deformations are presented in red and happen in the diaphragm near the heart area, which is in accordance with the large differences of the difference images in the first row.

Figure 5 shows the deformation field before and after refinement. The deformation field before refinement appears to be highly irregular, with visible artifacts and unrealistic deformations. In contrast, the deformation field after refinement is significantly smoother and more natural-looking, with many of the aforementioned artifacts and deformations corrected.

## 5. Discussion

This paper proposes a method for deformable registration of lung CT scans in the same patient, which combines large deformation decomposition and attention-guided refinement. The results obtained using this approach show that it outperforms other state-of-the-art methods on a publicly available dataset. The method’s ability to handle large deformations and variations in lung shape and texture is considered one of its key strengths. The use of attention mechanisms allows the method to focus on the most relevant features for registration and ignore noisy or irrelevant features, thereby increasing the algorithm’s robustness and accuracy even in challenging cases.

Looking to the future, there are several avenues for further research that we intend to pursue. One important area for improvement is the addition of boundary constraints to our registration algorithm. We observed some boundary problems in the deformation field. In the future, we will solve this problem by adding a boundary constraint or applying a lung mask during registration. By incorporating boundary constraints, we can ensure that our registration method is more robust to variations in lung shape and better able to handle boundary artifacts that can arise in CT images.

In addition to boundary constraints, we also plan to explore other regularization techniques that may further improve the accuracy and robustness of our registration method. One promising approach is to incorporate information about lung function, such as pulmonary ventilation or perfusion, into our registration algorithm. This could enable us to better account for physiological variations between patients and to tailor the registration process to individual patient needs.

In addition to further improving our intra-patient lung CT registration algorithm, we also plan to evaluate its performance on other datasets and modalities. This will enable us to test the robustness of our approach under different conditions and to gain a better understanding of its potential applications in a variety of clinical settings. One of our future plans is to test the performance of our algorithm on datasets containing patients with diverse lung diseases such as lung cancer or chronic obstructive pulmonary disease (COPD). By testing our method on a broader range of patient populations, we can better assess its ability to handle variations in lung shape and texture that may be present in different disease states.

## 6. Conclusions

This paper explores a straightforward and efficient approach for learning the large deformation field in lung CT image registration, which is essential for image-guided navigation systems. The registration accuracy is improved by breaking down the large deformation field into smaller intermediate fields and then combining them through a composition process followed by refinement using an attention layer. The experimental results demonstrate that our method outperforms existing methods. Future work will involve investigating boundary constraints and regularizations, as well as conducting a large-scale evaluation of different imaging modalities.

## Figures and Tables

**Figure 1 bioengineering-10-00562-f001:**
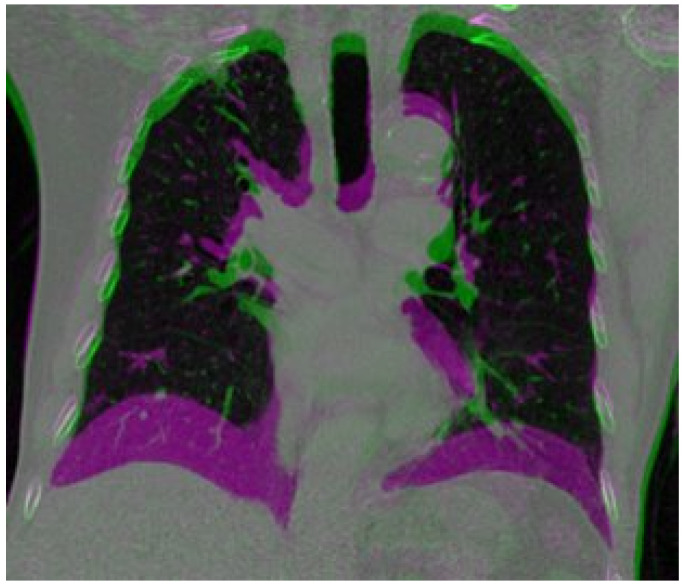
A composite image by overlaying the inspiration image onto the expiration image. The large differences are presented as magenta color.

**Figure 2 bioengineering-10-00562-f002:**
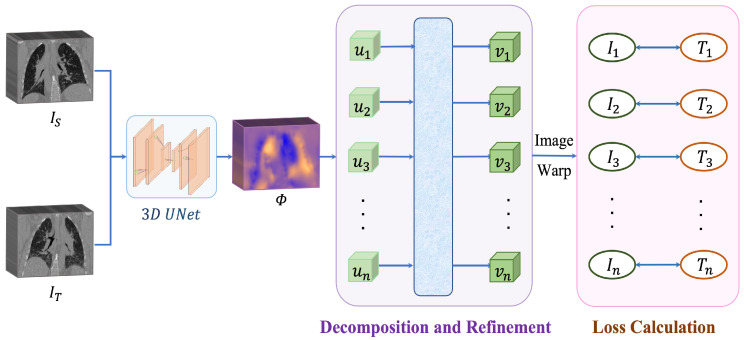
The illustrative pipeline of our method. The 3D Unet takes the source image and the target image as inputs and outputs a deformation field. Then, the deformation field is decomposed into several intermediate deformation fields, these deformation fields are refined through the attention layer. Finally, the network is updated by minimizing the similarity losses.

**Figure 3 bioengineering-10-00562-f003:**
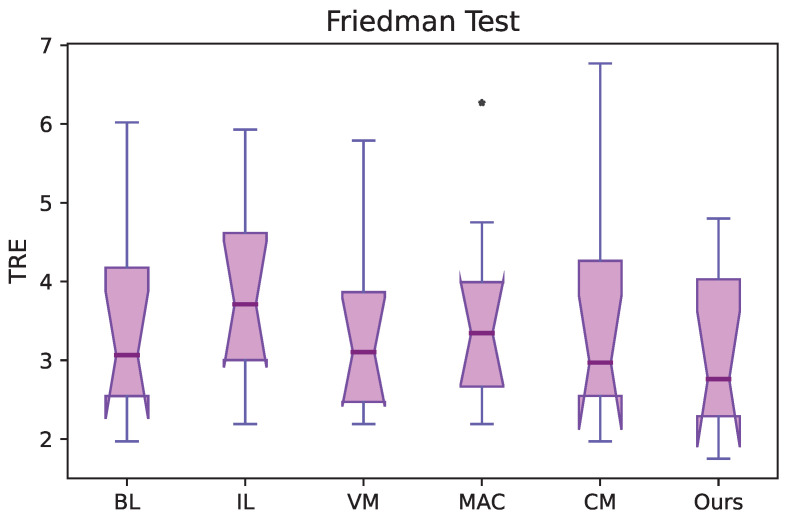
Friedman test. With $\chi_{Friedman}^{2}\left(5\right)=48.37$ and $p=2.2\times 10^{-7}$, we can conclude that our method has significant improvement over other algorithms.

**Figure 4 bioengineering-10-00562-f004:**
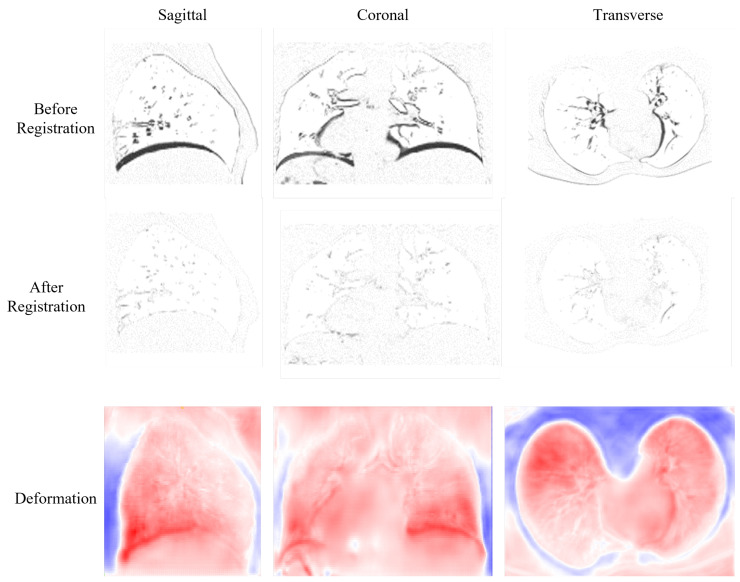
Visualization results for the difference images before and after registration, and deformation field from three perspectives.

**Figure 5 bioengineering-10-00562-f005:**
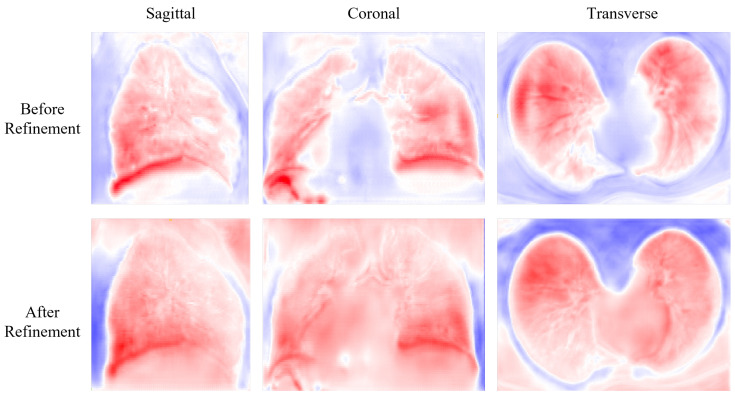
Visualization results for the slides of the deformation field before and after refinement.

**Table 1 bioengineering-10-00562-t001:** Numerical comparison of our proposed framework vs. other existing image registration algorithms. The numerical values reflect the TRE metric (mm) for the DIRLab 4DCT dataset. The best results are highlighted in bold font.

4DCT	BL	IL	VM	MAC	CM	Ours
01	2.29	4.24	2.28	2.35	**2.01**	2.20
02	1.97	2.19	2.19	2.19	1.97	**1.75**
03	2.62	2.52	2.43	2.68	2.98	**2.18**
04	2.70	3.16	2.59	2.66	**2.48**	2.56
05	4.16	4.74	**3.90**	4.12	4.03	4.45
06	**2.52**	3.18	3.14	3.32	2.75	2.64
07	4.18	4.24	3.76	3.61	4.34	**3.39**
08	6.02	5.93	5.79	6.27	6.77	**4.24**
09	3.43	2.95	3.07	3.37	2.96	**2.88**
10	5.40	5.36	4.69	**4.75**	5.35	4.80
*Avg.*	3.53	3.85	3.38	3.53	3.56	**3.11**
*Std.*	1.38	1.25	1.17	1.25	1.56	**1.06**

**Table 2 bioengineering-10-00562-t002:** Comparison of our proposed method vs. other existing image registration algorithms on the percent of negative values of Jacobian determinant. The best results are highlighted in bold font.

Methods	BL	IL	VM	MAC	CM	Ours
|J|<0(%)	0.023	0.018	0.020	0.020	0.022	**0.016**

## Data Availability

The data is publically available and the link is provided in Section 3.1.

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
