# Peer review of "Intra-Patient Lung CT Registration through Large Deformation Decomposition and Attention-Guided Refinement"

_bioengineering, 2023, doi:10.3390/bioengineering10050562_

Round 1

Reviewer 1 Report

The paper demonstrates the advantage of a learning (3D U-Net) - based registration approach, here including additionally decomposition, refinement and self-attention steps, applied for deformable medical image registration of CT volume images of lung tissue data sets.

The contribution is well written and consistent in its results. Comparisons with other relevant registration methods are shown, statistical analysis is performed, and it seems that the suggested approach is helpful for achieving almost a better registration performance.

Some questions/ information could be more discussed/provided:

·        Computational load/time due to the decomposition and refinement steps.

·        The deformation function Φ seems to be always chosen the same for all time steps; but they could also be adapted for the course of the decompositions.

·        How the spatial information is included in detail?

Check the text for smaller typos, e.g. 2,5 vs. 2.5; at l. 234

Fig.4 black/white images: Check if display in inverted gray scale or log scale of intensity gives better visibility of depicted differences. In a printed version it is almost not visible currently.

Author Response

Thank you for the detailed feedback on our paper. We appreciate the positive evaluation of our approach and the suggestions for improvement.

Regarding the computational load, we acknowledge that the decomposition and refinement steps do add to the overall computational complexity of our approach. For the decomposition and refinement method, extra 9 GB memory are needed for training the model. The proposed method did not result in a significant increase in computational time.

The deformation field Φ is updated after every iteration, and in each iteration, the decomposition is adapted by multiplying a coefficient to the deformation field.

Regarding the inclusion of spatial information, all spatial information is included in the deformation field, every vector in the deformation field corresponds to the displacement of a point in one image to its corresponding point in the other image.

Thank you for pointing out the typo at l. 234; we corrected this in the revised version.

We appreciate the suggestion to explore different visualization options for the black/white images in Fig.4 and displayed the results with inverted gray scale to enhance visibility.

Thank you for your valuable comments.

Reviewer 2 Report

I am glad to review the manuscript entitled Intra-patient lung CT registration through large deformation decomposition and attention-guided refinements. The research is well conducted and the results are promising, and I feel like it could be accepted, although some suggestions may render it even better. In this manuscript, the authors proposed a simple yet effective strategy to improve the register performance of the lung CT images. The deformation field Φ is first generated through a 3D UNet, and then decomposed into a linear interpolated series of intermediate fields ui. The intermediate fields
are then refined through the self-attention mechanism. The performance was validated using a public dataset with landmarks.
The manuscript is very well written, clearly presenting the superiority of the proposed method with a solid quantitative and qualitative metrics. However, the authors are strongly encouraged to conduct some further validation experiments to fully explore the strategy proposed.
1. Although the current strategy is specifically designed for lung CT registration, I am wondering if it is possible to apply it to other modalities, such as the dataset used in your reference 17. It would be very convincing if the proposed strategy can be generalized to other modalities such as PET or contrast-enhanced CT.
2. Is it possible to visualize the pristine and the refined deformation fields? I think it would be interesting to see how the attention-guided refinement works.
3. The authors are encouraged to conduct some ablation studies, such as removing the attention-guided refinement, or changing the step size n, or altering the attention layer, or applying some other backbones, to see how the performance is affected.
Thank you!

Author Response

Thank you for your thorough and constructive review of our manuscript. We appreciate your positive comments and suggestions for further improvements.

We agree that it would be interesting to investigate the generalizability of our proposed method to other modalities such as PET or contrast-enhanced CT. We will consider including such experiments in future work to demonstrate the versatility of our approach.

We apologize for not including the visualization of the pristine and refined deformation fields in the current manuscript. We agree that it would be informative to include these visualizations to show the effectiveness of the attention-guided refinement mechanism. We added these visualizations (Fig. 5) in the revised manuscript.

We appreciate your suggestion to conduct ablation studies to analyze the impact of different components of our method. 1) After removing the attention layer, the model achieved a score of 3.53. However, with the inclusion of the attentional layer, the model was able to achieve a better score of 3.11. The result with attention layer is better. 2) Change the step size n, we set n as 10 for an ablation, the result is 3.19. 3) Alter the attention layer, we tried a different self-attention structure by adjusting the residual layer before the layer norm, the result is 3.642, which is worse than original attentional layer.  

Once again, we thank you for your valuable feedback and comments, which we carefully considered in revising the manuscript.